# The Role of Endoscopy in the Palliation of Pancreatico-Biliary Cancers: Biliary Drainage, Management of Gastrointestinal Obstruction, and Role in Relief of Oncologic Pain

**DOI:** 10.3390/cancers15225367

**Published:** 2023-11-10

**Authors:** Giacomo Emanuele Maria Rizzo, Lucio Carrozza, Gabriele Rancatore, Cecilia Binda, Carlo Fabbri, Andrea Anderloni, Ilaria Tarantino

**Affiliations:** 1Endoscopy Unit, Department of Diagnostic and Therapeutic Services, IRCCS-ISMETT Palermo, 90127 Palermo, Italy; grizzo@ismett.edu (G.E.M.R.); lcarrozza@ismett.edu (L.C.); grancatore@ismett.edu (G.R.); 2Ph.D. Program, Department of Surgical, Oncological and Oral Sciences (Di.Chir.On.S.), University of Palermo, 90133 Palermo, Italy; 3Gastroenterology and Digestive Endoscopy Unit, Forlì-Cesena Hospitals, AUSL Romagna, 48100 Forlì-Cesena, Italy; cecilia.binda@gmail.com (C.B.); carlo.fabbri@auslromagna.it (C.F.); 4Gastroenterology and Digestive Endoscopy Unit, Fondazione I.R.C.C.S. Policlinico San Matteo, Viale Camillo Golgi 19, 27100 Pavia, Italy; andrea_anderloni@hotmail.com

**Keywords:** palliation, biliopancreatic cancer, endoscopy, biliary obstruction, pain, oncology

## Abstract

**Simple Summary:**

Palliative endoscopy has a fundamental role in the management of patients with advanced bilio-pancreatic cancers, which can involve the biliary tract and infiltrate the duodenal lumen or other close organs. Clinical presentations of these advanced cancers are mainly gastric outlet obstruction (GOO), obstructive jaundice, and unresponsive pain, which influence the patient’s quality of life (QoL) and the oncologic management in terms of initiating or restarting systemic therapy. Our aim was to perform a literature review focusing on the role of endoscopy in the palliation of these advanced pancreatic and biliary cancers.

**Abstract:**

Therapeutic endoscopy permits many and various treatments for cancer palliation in patients with bilio-pancreatic cancers, enabling different options, supporting patients during their route to oncologic treatments, and trying to improve their quality of life. Therefore, both endoscopic and endoscopic ultrasound (EUS)-guided techniques are performed in this scenario. We performed a literature review focusing on the role of endoscopy in the palliation of those advanced pancreatic and biliary cancers developing malignant biliary obstruction (MBO), gastric outlet obstruction (GOO), and pain unresponsive to medical therapies. Therefore, we explored and focused on the clinical outcomes of endoscopic procedures in this scenario. In fact, the endoscopic treatment is based on achieving biliary drainage in the case of MBO through endoscopic retrograde cholangiopancreatography (ERCP) or EUS-guided biliary drainage (EUS-BD), while GOO is endoscopically treated through the deployment of an enteral stent or the creation of EUS-guided gastro-entero-anastomosis (EUS-GEA). Furthermore, untreatable chronic abdominal pain is a major issue in patients unresponsive to high doses of painkillers, so EUS-guided celiac plexus neurolysis (CPN) or celiac ganglia neurolysis (CGN) helps to reduce dosage and have better pain control. Therefore, therapeutic endoscopy in the palliative setting is an effective and safe approach for managing most of the clinical manifestations of advanced biliopancreatic tumors.

## 1. Introduction

Endoscopy is the standard of care for the palliation of advanced cancers involving the gastrointestinal (GI) tract. The role of palliative endoscopy is variable and dependent on cancer advancement, which moves from the involvement of the biliary tract to the infiltration of the duodenal lumen or other close organs. Therapeutic endoscopy, including endoscopic ultrasound (EUS), has improved over the years to overcome the clinical symptoms of advanced neoplastic diseases, permitting different options, supporting patients during their route toward starting systemic chemotherapy, and even trying to improve their quality of life (QoL) [1]. 

We aimed to perform a literature review focusing on the role of endoscopy in the palliation of advanced pancreatic-biliary cancers, in order to highlight the technical and clinical aspects of those endoscopic procedures which are strengthening as first-line approaches in the case of cancer palliation.

## 2. Clinical Aspects of Advanced Pancreatic-Biliary Cancer

Pancreatic and biliary cancers are among the most aggressive cancers [2]. In the United States (US), researchers have estimated the average annual incidence rate (2015–2019) of pancreatic cancers at 13.2 per 100,000 inhabitants [2], so estimated new cases and deaths are 64,050 and 50,550 in 2023 [3]. Annual new cases of gallbladder and other biliary cancers, indeed, are estimated to be 12,220 in the US, while estimated deaths are 4510 [3]. Surely, even metastasis and neoplastic lymph nodes may involve the biliary tract and duodenal lumen, [4] creating the need for endoscopic treatments. Nowadays, the 5-year survival rate at the time of diagnosis is still dramatically low, being 10% for pancreatic cancer and 18% for localized/regional extrahepatic bile duct cancers (both hilar and distal) in the USA [3,5]. Therefore, palliation is the main aim in those advanced cases developing jaundice, oncologic pain, or vomiting, so the management of the latter conditions becomes of primary relevance. The trigger for mechanical obstruction is usually an infiltration or compression of the biliary and duodenal tract by the malignancy (Figure 1), which then clinically produces malignant biliary obstruction (MBO) or gastric outlet obstruction (GOO). On the other hand, both the malignancy itself and the involvement of nerves cause severe oncologic pain, which is arduous to resolve with a single intervention such as painkiller administration; nonetheless, alternative endoscopic therapies targeting the celiac plexus are available (Figure 1) [6]. However, MBO and GOO can be endoscopically treated, being caused by a mechanical obstruction, while cancer pain needs to be first treated by an expert in the field of pain therapy, even if EUS-guided therapeutic options may complement medical therapies [7,8].

### Materials and Methods

This is a comprehensive review of the role of endoscopy in the palliation of advanced pancreatic–biliary cancers. Considering the vastness of the topic, the search strategy, materials, and methods were adapted to each main topic of the review, and they are more deeply discussed in the Appendix A [9,10]. Generally, the identification of the literature, the selection of sources, and the analysis, synthesis, and organization of the information were conducted by three researchers (G.E.M.R., L.C. and G.R.).

## 3. Endoscopic Treatments

Palliative endoscopic treatments in this scenario include procedures involving both endoscopic and EUS-guided techniques depending on the aim of the treatment and location of the issue (e.g., drainage, anastomosis creation, alcohol injection, ablation, debulking, and so on). In the case of MBO, which is one of the most common complications of malignancies involving the hepato-biliary-pancreatic system, the endoscopic treatment is based on achieving biliary drainage through endoscopic retrograde cholangiopancreatography (ERCP) or EUS-guided biliary drainage (EUS-BD). The resolution of jaundice reduces the risk of cholangitis and sepsis, and consequently improves QoL [11]. Furthermore, MBO can be divided based on its location into malignant distal biliary obstruction (dMBO) and malignant proximal biliary obstruction (pMBO). In the case of GOO, endoscopic treatment can include either duodenal stenting or EUS-guided gastro-enteroanastomosis (EUS-GEA). On the other hand, intractable oncologic pain in the case of bilio-pancreatic malignancies has been treated through celiac plexus neurolysis (CPN) over the years, firstly percutaneously, then through an EUS-guided approach, showing similar effectiveness and safety in randomized trials [12].

### 3.1. Malignant Biliary Obstruction (MBO)

#### 3.1.1. Role of Endoscopic Retrograde Cholangiopancreatography (ERCP)

The transpapillary approach through ERCP is a milestone in the management of MBO with the advantages of avoiding external drainage [13], shorter hospitalization times, and lower rates of adverse events (8.6% vs. 12.3%, *p* < 0.001) compared to PTBD [14]. ERCP is also associated with lower rates of morbidity, peri- and post-procedural complications, and 30-day mortality (16.3% vs. 9.6%) when compared with the surgical approach, although surgical biliodigestive anastomosis showed a reduction in the rates of recurrent jaundice [15,16]. However, no interruption in the administration of oncological treatments is fundamental to achieving better oncological outcomes such as overall survival (OS) and progression-free survival (PFS), so the goal of ERCP is to permit BD in as many patients as possible.

#### 3.1.2. Distal Malignant Biliary Obstruction (dMBO)

DMBO refers to malignant involvement of the distal part of the common bile duct (CBD) and it may be caused by intrinsic or extrinsic compression such as pancreatic head cancer, cholangiocarcinoma, ampullary cancer, or compression of metastatic lymph nodes [4,17]. Endoscopic treatments were historically based on ERCP, which is still considered the gold standard, even if EUS-guided approaches, which were initially used after ERCP failure, are becoming an alternative primary treatment, as suggested by recent studies and ongoing trials [18,19,20]. The European Society of Gastrointestinal Endoscopy (ESGE) guidelines recommend biliary self-expandable metal stent (SEMS) insertion for palliative drainage [21]. The choice of the type of stent to use is influenced by several factors such as the location of the stenosis, the patient’s prognosis, and the availability of the prosthesis. There is enough evidence in the literature to suggest the choice of SEMS over a plastic prosthesis since remaining patent for longer improves patient outcomes. In the meta-analysis by Moole et al., where 11 studies with a total of 947 patients were selected, the pooled analysis of SEMS patency was 167 days, unlike the 73 days of the plastic stent [22]. Either covered SEMS (C-SEMS) or uncovered (U-SEMS) may be used, even if there is still a debate over which is the best due to conflicting results in the literature. In fact, C-SEMS seemed to prolong stent patency but had a higher migration rate [21] compared to U-SEMS, where tumor ingrowth through the metal mesh fixes the stent but reduces patency, even if a meta-analysis including nine randomized controlled trials (RCTs) found no difference in the length of stent patency [23]. Further meta-analyses evaluated the use of C-SEMS vs. U-SEMS without finding significant differences in clinical outcomes [24,25]. Regarding the safety and the rate of adverse events (AEs), the abovementioned meta-analysis did not demonstrate any higher risk of cholecystitis after C-SEMS insertion. Similarly, no differences in pancreatitis rate were shown between C-SEMS and U-SEMS. However, a novel type of stent was developed to counter stent ingrowth, the chemotherapy drug-eluting stent, but a meta-analysis of five studies comparing drug-eluting stents (197 patients) to SEMS (151 patients) reported a stent patency of 168 days vs. 149 days, respectively, with no major differences in the rates of cholecystitis (6.5% vs. 5.0%) or cholangitis (17% vs. 15%) [26]. Therefore, those stents have yet to receive receive FDA approval. Percutaneous biliary drainage (PTBD) has also been used as an alternative, showing similar efficacy with no significant differences in survival time or costs compared to endoscopic biliary drainage [27], but it needs an external approach and it could impact the QoL of patients. When jaundice secondary to pancreatic neoplasms is susceptible to neo-adjuvant chemotherapy, plastic biliary stent placement (of at least 10 Fr) was suggested until a few years ago, because the inflammatory reaction created by a SEMS made the surgical procedure more complex. A recent systematic review and meta-analysis by Du et al. conducted with the aim of comparing the clinical efficacy of metal stents versus plastic stents in patients undergoing neoadjuvant therapy included two randomized trials and six retrospective studies with a total of 316 patients, showing no significant differences in terms of operative and postoperative time, and the need for endoscopic reintervention and stent-related complications were significantly lower in the group treated with metal stents than in the one treated with plastic stents, respectively (18% vs. 80% and 15% vs. 44%) [28]. 

#### 3.1.3. Proximal Malignant Biliary Obstruction (pMBO)

PMBO refers to malignant involvement of the proximal part of CBD caused by intrinsic obstruction or extrinsic compression by cancers, and it can involve the confluence of the hepatic ducts, often called ‘Klatskin tumor’, causing a malignant hilar biliary obstruction (hMBO) (Figure 2) [29,30]. Therefore, biliary drainage of the Klatskin tumors is strongly influenced by the extension of the neoplastic tissue, well-differentiated by the Bismuth classification (Appendix A) [31], because of the lower probability of concurrently draining through ERCP all of the hepatic segments when approaching a Bismuth type IV or III [21].

The retrograde approach is sometimes not the best option in the case of pMBO, especially when the tumor involves biliary confluence into both the right and left biliary ducts (type IV according to the Bismuth classification), because in these difficult cases sometimes it is not possible to drain both of the ducts, so patients do not resolve jaundice. Therefore, in the case of Klatskin tumor Bismuth IV or III it is extremely important to have a multidisciplinary approach together with an interventional radiologist in order to drain all the segments through a rendezvous or with an additional insertion of a PTBD [21]. A systematic review and meta-analysis of nine studies (*n* = 546 patients) showed a higher success rate with PTBD than ERCP in types III/IV, with comparable rates of adverse events and 30-day mortality [32]. On the other hand, Inamdar et al. reported that biliary drainage through ERCP showed a lower adverse event rate and shorter hospitalization when compared with PTBD [14]. Moreover, in a propensity score matching analysis, patients who underwent PTBD had lower overall survival and a higher risk for seeding metastasis when compared with ERCP [33]. Generally, PTBD is preferred when a patient has an altered gastro-duodenal anatomy, when the bile ducts to be drained are not accessible by ERCP, or when ERCP does not achieve adequate biliary drainage. Regardless of the method used, achieving ≥50% of total liver volume drainage is essential to relieve jaundice and reduce the risk of cholangitis. This was associated with longer overall survival particularly in the Bismuth III type [34]. Similarly, in their retrospective study, Takahashi et al. [35] correlated the percentage of liver volume to be drained with the patient’s liver function and concluded that effective biliary drainage is achieved in patients with preserved liver function when >33% of the liver volume is drained, and in those with impaired liver function when >50% is drained. Anyway, regarding ERCP stenting, different meta-analyses comparing SEMS to plastic stents resulted in longer patient survival, lower risk of stent dysfunction and infection, and fewer reoperations when SEMS was deployed [36,37]. 

#### 3.1.4. Endoscopic Ultrasound Biliary Drainage (EUS-BD)

Although ERCP remains the gold standard in the treatment of dMBO, the international consensus statement for the management of malignant distal biliary stricture recommends that, when expertise is available, ultrasound endoscopic biliary drainage (EUS-BD) is an effective option in three situations: failed ERCP, difficult biliary cannulation, and postsurgical anatomy [13]. In fact, although PTBD has long been utilized, EUS-BD is a less invasive option with fewer procedure-related adverse events (8.80% vs. 31.22%, *p* = 0.022) and lower reintervention rates (0.34 vs. 0.93, *p* = 0.02) shown in a randomized open-label study [38], and recommended by European guidelines over PTBD [21]. Subsequently, these data were confirmed by a meta-analysis including 483 patients [39]. A systematic review of 42 studies including 1192 patients undergoing EUS-BD after ERCP failure reported a technical success rate of 94.7%, a clinical success rate of 91.6%, and an adverse event rate of 23%, which included bile leak (4.03%), bleeding (4.03%), pneumoperitoneum (3.02%), stent migration (2.68%), cholangitis (2.43%), abdominal pain (1.51%) and peritonitis (1.26%) [40]. Moreover, EUS-BD techniques can be divided according to the anatomical location and the puncture site of the biliary access into choledochoduodenostomy (CDS), hepaticogastrostomy (HGS), rendezvous technique (RV), antegrade biliary stenting (AG), and gallbladder drainage (GBD) (Figure 3). 

In patients in whom ERCP fails, endoscopic ultrasound-guided choledochoduodenostomy (EUS-CDS) is considered the preferred choice for dMBO [41], as confirmed in a multicenter retrospective study comparing EUS-CDS to PTBD and demonstrating higher clinical success (84.6% vs. 62.1%, *p* = 0,04) for EUS-CDS with a significantly lower rate of reoperation (10.7% vs. 77.6%, *p* < 0.001) [42]. Biliary drainage through EUS-CDS permits direct access to the CBD from the duodenum creating a choledochoduodenostomy through the deployment of a plastic stent or fully covered metal stent, which is extremely useful and successful in the case of dMBO. Initially, FC-SEMS were preferred over plastic stents for CDS, as they have significantly lower rates of adverse events (13.0% vs. 42.8%, *p* = 0.01) and better stent patency [43,44], even if FC-SEMS theoretically increase the risk of stent migration. In this context, a fully-covered short metal stent with double flanges (lumen-apposing metal stent, LAMS) was developed for EUS-guided procedures about a decade ago [45] and it is on its way to becoming the preferred choice in the case of EUS-CDS. Furthermore, the application of the electrocautery-enhanced tip of the LAMS catheter has enabled a “free-hand”, “single-step”, and “exchange-free” procedure, making direct organ access possible without using further devices such as needles, guidewires, or dilator devices. A systematic review and meta-analysis containing seven studies including 284 patients who underwent EUS-BD using LAMS after ERCP failure showed high technical and clinical success rates (95.7% and 95.9%, respectively) with a 5.2% pooled rate of post-procedural adverse events and an 8.7% rate of recurrence [46]. Finally, those results were confirmed by a recent large multicenter study [47]. However, no differences in the technical and clinical success or post-procedure-related adverse events comparing LAMS vs. SEMS have been found so far [48,49], even if nowadays experts seem to prefer LAMS over FC-SEMS. On the other hand, EUS-guided hepatogastrostomy (EUS-HGS) is preferred in the case of hMBO, because it permits the creation of a fistulous duct between the gastric wall and the left intrahepatic duct, unlike EUS-CDS, which is indicated in dMBO. Moreover, when ERCP and/or PTBD do not achieve clinical success with adequate biliary drainage, ESGE suggests EUS-guided biliary drainage with EUS-HGS only for malignant inoperable hilar biliary obstruction with a dilated left hepatic duct [50]. However, current data on which is the best choice for MBO are conflicting, with some reports showing higher safety for the transduodenal route, while others have shown no such difference [51,52]. In a small randomized study comparing 25 patients who received EUS-HGS and 24 who received EUS-CDS, the clinical success of EUS-HGS was higher (91% versus 77%); however, adverse events were also slightly higher (20% vs. 12.5%), although neither outcome reached statistical significance [53]. A systematic review and meta-analysis of 10 studies by Uemura et al. comparing EUS-HGS (*n* = 208) and EUS-CDS (*n* = 226) found no difference in technical success (94.1% vs. 93.7%), clinical success (88.5% vs. 84.5%), or rates of adverse events [54]. Furthermore, a multicenter study on long-term patency of the two techniques conducted on 182 patients (95 EUS-HGS vs. 87 EUS-CDS) showed that EUS-CDS was associated with being 4.5 times more likely to achieve longer stent patency at the expense of a higher rate of adverse events [55]. Moreover, the EUS-guided gallbladder drainage appears a valid alternative as a rescue treatment after ERCP and EUS-CDS failure, showing adequate efficacy and safety for those patients who have dMBO and no involvement of the cystic duct. Therefore, a recent multicenter study involving 48 patients showed 100% and 81.3% technical and clinical success rates, respectively, with 10.4% of AEs [56]. Thus, the choice between these approaches is based on a combination of factors including procedural proficiency, risk of adverse events, and anatomical factors, such as the presence of a dilated bile duct or bile radicals, duodenal stenosis, and altered anatomy [57].

#### 3.1.5. Comparison between ERCP and EUS-BD

The first study that compared ERPC vs. EUS in the drainage of biliary obstruction was a multicenter retrospective study demonstrating similar rates of technical success (94.23% for ERCP vs. 93.26% for EUS-BD, *p* = 1.00) and adverse events (8.65% for ERCP vs. 8.65% for EUS-BD); however, the ERCP was burdened by 4.8% of post-procedural pancreatitis [58]. Similar results were found in a meta-analysis showing that both techniques were equally effective in achieving biliary drainage (ERCP = 94.73%; EUS = 93.67; pooled odds ratio (OR): 1.20; 95% confidence interval (CI): 0.44–3.24) while there was no significant difference in adverse events (ERCP = 22.3%; EUS = 15.2%; OR 1.59; 95% CI 0.89–2.84), and furthermore, post-procedure pancreatitis (PEP) was significantly higher for ERCP (9.5% vs. EUS = 0; risk difference: 8%; 95% CI: 1–14%) [59]. Additionally, in cases of a gastroduodenal stent, the EUS-guided approach has been proven as technically and clinically superior when compared to ERCP [60], especially in the setting of concomitant double obstruction [61]. Finally, another systematic review and meta-analysis confirmed no significant differences in technical and clinical success between ERCP and EUS-BD, with lower rates of reintervention for EUS-BD [62].

### 3.2. Malignant Gastric Outlet Obstruction (mGOO)

The most frequent cause of mGOO in Western countries is pancreatic adenocarcinoma (between 15 and 25% of patients with pancreatic cancer develop MGOO during the course of the disease) [63]. Anyway, any other neoplasia occluding pylori or duodenum leads to mGOO, even if less frequently, as in the case of gastric cancer, neoplasms of the proximal duodenum and ampulla, local extension of advanced gallbladder carcinoma or cholangiocarcinoma, metastatic or primary malignancy in the duodenum, gastric carcinoid, or gastrointestinal stromal tumors/gastric leiomyosarcomas. The GOO-related clinical manifestations include abdominal pain, nausea and/or vomiting, early satiety and/or anorexia, bloating, and weight loss, which in the long term lead to cachexia. Furthermore, cancer progression increases these symptoms, also leading to dystrophy, general fatigue, dehydration, and electrolyte balance disorders [64,65]. By the way, the prognosis of these patients is related to tumor progression or an impaired general condition, so patient survival is also closely associated with the development of cachexia [66]. However, the lack of minimally invasive treatments in the past caused those patients with mGOO to undergo surgery to bypass the GI obstruction through a gastrojejunostomy, which was associated with a biliary shunt when occurring concurrently with biliary obstruction. However, those patients with advanced disease involving the GI tract are usually in poor condition and are not good candidates for surgery, so less invasive treatments have been developed over the years to rapidly and more safely treat this condition, improving many consequent outcomes, such as time to re-feeding, hospitalization time, and management costs. This goal was achieved with the development of endoscopic approaches such as enteral stent placement and more recently the creation of EUS-guided gastro-entero-anastomosis (EUS-GEA). Moreover, GOO-related symptoms were gathered into a score by Adler and colleagues, the gastric outlet obstruction scoring system (GOOSS score, Table 1), which is extremely helpful in easily following clinical outcomes after procedures through the improvement of patients’ feeding [67]. In fact, clinical success is generally defined by remission of obstructive symptoms and resumption of oral feeding, when treating mGOO. Anyway, the choice between GEA and enteral stenting is dependent on different variables, so a prognostic scoring system was recently developed for patients with MGOO due to pancreatic adenocarcinoma in order to propose the best procedure depending on the survival predicted: a score between 0 and 1 indicates a better prognosis for the patient, so GEA should be preferred, while patients with a score between 2 and 4 have a worse prognosis and enteral stenting could be a better option [68].

#### 3.2.1. Enteral Stenting

Endoscopic placement of an enteral stent was the first endoscopic option for treating mGOO as an alternative to surgical GEA [69], being extremely useful for those patients unfit for surgery, but it had a high rate of reintervention and low patency time compared to gastrojejunostomy [70]. Anyway, enteral stenting is alternatively used to re-establish channeling in patients with malignant gastrointestinal obstruction who are not eligible for surgery and with short life expectancy (less than 6 months) [71]. The first case reported in the literature of a self-expanding metallic stent for GOO dates back to 1992 [69]. Various studies have supported the efficacy and safety of enteral stenting in the management of unresectable mGOO since then [72,73,74]. Technical success, defined as the correct placement of the stent across the tumor stenosis, is frequently very high. In a systematic review with pooled analysis including 19 studies and 1281 patients, the overall pooled technical success rate was 97.3% and the clinical success rate was 85.7% [75]. According to the technique, a guidewire is placed beyond the duodenal stenosis over which the stent is then slid under radiological and endoscopic view (through-the-scope techniques). Finally, the injection of intraluminal contrast dye verifies both the regular flow through the SEMS after the obstruction site and the absence of any extra-luminal diffusion. Currently, we have three main types of enteral self-expandable metal stents (uncovered, partially covered, or fully covered) with different lengths, diameters, and radial expansive forces. In a systematic review including five trials with a total of 443 patients with MGOO, the authors compared the outcomes of covered SEMS vs. uncovered SEMS, showing that covered SEMS had a lower rate of stent occlusion (number-needed-to-treat, NNT, of 5) despite higher rates of stent migration compared with the uncovered SEMS (RD: 0.09, 95% CI [0.04, 0.14], I^2^ 9%, with a number-necessary-to-harm [NNH] of 11) [76]. In 2018, a systematic review confirmed that duodenal stenting had a faster return to oral intake, and shorter hospitalization time despite an increased recurrence of symptoms and increased reintervention rate when compared to surgical GEA [77]. As far as adverse events are concerned, the percentage varies between 0 and 30%, and they are strictly connected to the definition indicated in the study. Therefore, we have minor adverse events such as mild pain, nausea, and vomiting and major adverse events such as bleeding, perforation, and stent migration [78]. In the particular case where patients develop secondary MGOO and/or concurrent biliary obstruction, the positioning of the SEMS may increase the risk of biliary dysfunction. In the analysis by Hamada T et al., 410 patients with distal malignant biliary obstruction were enrolled and a duodenal SEMS was positioned in 33 (8%), 17 (52%) of whom developed biliary dysfunction with an average of 64 days after stent placement [79]. 

#### 3.2.2. EUS-Guided Gastro-Entero-Anastomosis

EUS-guided GEA represents a novel and minimally invasive alternative to surgery and enteral stent for managing malignant GOO, and the literature shows increasing evidence in support of the advantages of EUS-guided anastomoses. In the past, surgery for gastrojejunostomy bypass was the most common option [80], but nowadays the development of EUS-GEA permits a less invasive option with similar efficacy and either fewer days of hospitalization or time to oral feeding for creating a GEA. Moreover, when malignancy causes concurrent biliary and duodenal obstruction, the EUS-guided approach may become the preferred one in the current era of EUS-guided procedures, [61] even if depending on the location of the obstruction, as indicated by the “bilioduodenal” classification [81]. The first EUS-guided method to create a GEA was reported in 2003 [82] in a porcine model but without adequate devices. Nowadays, the number of devices has increased but the technique is still not completely standardized, so some dedicated groups have worked on recognizing the differences among techniques and tertiary centers in order to better understand which is the best approach [83]. In general, the endosonographer firstly advances a catheter (or a double balloon/single balloon enteric tube) over a stiff guidewire through the gastric or duodenal stricture, and then saline is injected downstream of the stricture in order to fill the jejunal lumen. Finally, after EUS-identification of the enlarged enteral loop (“target”), the distal flange of the LAMS is deployed into the jejunal lumen (using the hands-free technique or through a guidewire previously placed through loop puncture with a fine-needle) and the proximal flange is deployed into the gastric lumen (with or without the intra-channel release technique). As described above, different variants of the technique have been developed over the years, changing the devices used for GEA creation or for loop enlargement, or in techniques for the target loop puncture. By the way, EUS-GEA for the treatment of gastric outlet obstruction (GOO) was initially performed only with one type of electrocautery lumen-apposing metal stents (EC-LAMS), especially thanks to the releasing system permitting the easy use of the wireless “free-hand” technique, but the use of another EC-LAMS was recently reported in the creation of a EUS-GEA [84,85], so further comparisons are expected in the near future. In general, therefore, the techniques for EUS-GEA can be summarized as direct EUS-GE, balloon-assisted EUS-GE, EUS-guided double-balloon-occluded gastrojejunostomy bypass (EPASS), and the wireless EUS-guided gastroenterostomy simplified technique (WEST), techniques that are described in-depth in other technical studies [85,86,87,88,89,90]. All in all, EUS-GEA is changing the approach to mGOO, moving toward becoming the standard of care in the future. In a meta-analysis including twelve studies and 290 patients the pooled technical success rate was 93.5% (95% CI, 89.7–6.0%; I^2^ 0%) and the pooled clinical success rate was 90.1% (95% CI 85.5–93.4%; I^2^ 0%), even if the studies included different techniques, mostly direct EUS-GE (68.2%), and indications were for mGOO only in 62.4% of cases [91]. Recently, a further and updated meta-analysis including 1493 patients with both benign and malignant GOO treated with EUS-GEA showed technical success and clinical success rates of 94% and 89.9%, respectively. Furthermore, safety analysis showed a pooled rate of AEs of 13.1% [92]. Moreover, a recent multicenter retrospective study evaluated differences in treating mGOO with EUS-GEA (*n* = 187) vs. surgical gastrojejunostomy (SGJ, *n* = 123), showing significantly lower time to resumption of oral intake (1.40 vs. 4.06 days, *p* < 0.001) and a shorter length of stay (5.31 vs. 8.54 days, *p* < 0.001) comparing EUS-GEA with SGJ, with no differences in technical and clinical success between procedures (97.9% vs. 100% for TS and 94.1% vs. 94.3% for CS, respectively) [93]. In a matched comparison analysis of EUS-GEA vs. endoscopic stenting (ES), clinical success was, respectively, 100% vs. 75% (*p* = 0.006), with a lower recurrence rate (3.7% vs. 33.3%, *p* = 0.02) and a trend toward shorter time to chemotherapy [94]. However, a challenging scenario recently explored was the creation of EUS-GEA for GOO with peritoneal carcinomatosis, which showed slightly better outcomes when compared to SGJ, having comparable technical success (both 100%) and clinical success (88% vs. 85%, *p* > 0.99), but a lower rate of AEs (8% vs. 41%, *p* = 0.01, respectively). EUS-GEA is generally a safe technique, showing 12.9% of AEs in a prospective study evaluating 104 patients [94], which was similar to those pooled rates presented in the meta-analyses (13.1%), which was significantly low when compared to SGJ (13.4% vs. 33.3%, *p* < 0.001) [92,93]. Various comparative studies have been published in order to evaluate which is the most effective treatment for those patients developing mGOO, even if most of them have been retrospective so far. A recent meta-analysis of fifteen studies (*n* = 1441) showed higher pooled clinical success without recurrent GOO of EUS-GE when compared to ES or SGJ combined (OR, 2.60; 95% CI, 1.58–4.28) [95]. An overview of the outcomes, when comparing different techniques for mGOO, is shown in Table 2. Regarding the safety of these procedures, the table clearly shows some differences depending on the procedure, highlighting a generally better profile of the EUS-GE compared to a surgical approach, but even when compared to enteral stenting. Miller et al. showed a significantly lower pooled rate of AEs for the EUS-GE group compared to ES or SGJ grouped together (OR: 0.34; 95% CI: 0.20–0.58), or SGJ alone (OR: 0.17; 95% CI: 0.10–0.30) and no significant differences when compared to ES alone (OR: 0.57; 95% CI: 0.29–1.14) [95]. However, it is important to keep in mind that training for performing EUS-GE is still not well-established and these procedures are mainly performed by skilled and expert endosonographers, so this could be a bias in the context of real-world clinical practice.

#### 3.2.3. Natural Orifice Transluminal Endoscopic Surgery (NOTES)

Another endoscopic approach for palliation of tumors causing mGOO is the natural orifice transluminal endoscopic surgery (NOTES) for creating gastro-entero-anastomosis, which is still under development. Even if it has been proven to be as effective mostly in porcine models [105,106], it is still an option as a rescue therapy in the case of complete stent misdeployment during EUS-GEA [107,108]. Endoscopic access to the peritoneum was first described by Kalloo et al. [109] in a porcine model in 2004, changing our way of thinking about endoscopy and leading to the development of a new technique for creating EUS-GE or performing submucosal tunneling endoscopy. Various studies, mostly performed on animal models, demonstrated the feasibility and safety of performing NOTES-GE [106,110,111]. NOTES-GE includes several variations of the technique, but it generally starts in a similar way, with the identification of the small bowel segment closest to the gastric wall (usually corresponding to the ligament of Treitz) [112] through an echoendoscope. Then, a 19-gauge needle is used to insert a guidewire into the peritoneal space toward the ligament of Treitz, so the dilation of the tract permits the passage of a double channel forward-viewing endoscope into the peritoneal space. Under a direct endoscopic view, the small bowel distal to the obstruction is grasped by forceps while a 19-gauge needle punctures the small bowel inserting a guidewire. Therefore, an EC-LAMS is inserted under direct endoscopic visualization over the wire into the small bowel lumen, where the distal flange is deployed. Both the stent delivery catheter and the scope are finally pulled back within the gastric lumen, where the proximal flange is deployed creating the gastroenterostomy tract. Therefore, although animal data, as abovementioned, have demonstrated the feasibility of NOTES-GE, data on clinical settings are limited, and so the sample size is extremely small to apply it in clinical practice, despite the high technical and clinical success rates achieved in the described cases [112,113,114].

## 4. Pain Secondary to Bilio-Pancreatic Cancers

Patients with advanced bilio-pancreatic cancers may develop untreatable chronic abdominal pain, mainly due to the perineural invasion of tumor cells, and pain is present in 70–90% at diagnosis [115]. Pain management usually begins with medication titration in these oncological cases (i.e., progressing from nonsteroidal anti-inflammatory drugs to narcotics) but, unfortunately, they often are not able to fully relieve it despite adherence to the World Health Organization (WHO) analgesic ladder [116]. Moreover, celiac plexus neurolysis (CPN) also has a role in pain management in patients with advanced pancreatic cancer; in fact 16 trials have been published since 1997 evaluating its effectiveness in pain management and more than 50% of the patients enrolled had a reduction in pain intensity or decreased opioid consumption [117]. Therefore, alternative and additional therapeutic options to painkillers and opioids have been evaluated over the years, such as celiac plexus neurolysis (CPN) or celiac ganglia neurolysis (CGN) with various agents, administered either percutaneously or transgastrically [118]. CPN is the most widely used interventional procedure for the treatment of abdominal cancer pain, demonstrating efficacy for patients with both malignant and chronic non-malignant pain [119,120]. The celiac plexus is a dense network of autonomic fibers innervating visceral abdominal organs converging into the celiac ganglia, which are located in the retroperitoneum and adjacent to the origin of the celiac trunk. CPN may be able to reduce pain intensity and thus decrease systemic analgesic intake. Some authors have shown long-lasting pain relief for patients with pancreatic and intra-abdominal cancers with a benefit ranging from 50 days up to the time of death [121,122]. However, EUS technically permits performing CPN through the gastric wall, which allows for a safer and more effective procedure, as first described by Wiersema in 1996 and showing pain improvement in 79–88% of patients [123]. The safety profile is fundamental, because the EUS-guided transgastric approach allows direct access to the celiac plexus, leading to a reduction in the risk of injuries to the spinal nerve, diaphragm, or spinal artery. 

### EUS-Guided Neurolysis

Injection of substances into the celiac plexus is an established method for relieving pain in upper abdominal malignancies [124]. Absolute ethanol is commonly used after an injection of bupivacaine for performing CPN, while a combination of bupivacaine and triamcinolone is used in case of celiac plexus blockade. The safety of a combination of receiving 20 mL of 0.75% bupivacaine followed by 10 mL or 20 mL of alcohol for EUS-CPN was prospectively demonstrated in a cohort of 20 patients [125]. No major complications were seen in either group while minor self-limited AEs were seen in six (30%) subjects, including lightheadedness (5%), transient diarrhea (10%), and transient nausea and vomiting.

Technically, EUS-guided CPN consists of directly injecting substances in the two sides of the aorta at the level of origin of the celiac artery where the celiac ganglia are located, while maintaining the sagittal imaging of the aorta. Some authors inject 3 mL of 0.25% preservative-free bupivacaine followed by 10 mL of dehydrated 98% absolute ethanol into each side [126]. The result of the alcohol injection is an echogenic “cloud”, which may cause discomfort after the procedure. In 2001, Gunaratnam et al. [127]. performed EUS-CPN in 58 patients with pancreatic cancer pain, reporting pain relief in 78% of them. Further initial data showed low efficacy (68.1% of patients with pain relief [128]), so predictive factors were also explored in order to enable rational selection of the therapeutic strategy. Therefore, in the first analysis in 2011, direct invasion of the celiac plexus and left-sided distribution of the injected ethanol were identified as significant predictors of a negative response to CPN [128]. Another evaluation of predictive factors in 2021 confirmed celiac plexus invasion (13.2 OR, 95% CI 3.02–46.27, *p* = 0.003) as significant negative independent pain response factors to EUS-CPN, also adding invisible ganglia (49 OR, 95% CI 2.25–17.91, *p* = 0.011) and presence of distant metastases (6.84 OR, 95% CI 2.34–19.15, *p* = 0.022) [129]. In 2008, a meta-analysis including eight studies with 283 oncologic patients undergoing EUS-CPN showed a pooled proportion of pain relief of 80.12% (95% CI 74.47–85.22) [130]. However, a meta-analysis evaluating the bilateral and unilateral EUS-CPN approaches and including 437 patients did not find a significant difference between the two approaches both in terms of short-term pain relief (SMD = 0.31, 95% CI (−0.20, 0.81), *p* = 0.23) and response to treatment (RR = 0.99, 95% CI (0.77, 1.41), *p* = 0.97), even if only the bilateral approach showed a significant reduction in the postoperative use of analgesics (RR = 0.66, 95% CI (0.47, 0.94), *p* = 0.02) compared to the unilateral approach [131]. However, on the other hand, a more specific variant of neurolysis consists of directly injecting agents into the ganglia, which are visualized as small and hypoechoic oval images at EUS-view. One of the first studies performing EUS-CGN with alcohol in 17 patients with pancreatic cancer and in 5 patients with chronic pancreatitis resulted in an improvement of pain scores in 94% and 80% of patients, respectively [132]. Later, a multicenter randomized trial comparing EUS-CGN and EUS-CPN showed a higher positive response rate at 7 postoperative days (POD) in the CGN group (73.5%) than in the CPN group (45.5%; *p* = 0.026), confirmed when evaluating the complete response rate (CGN group 50.0% vs. CPN group 18.2%; *p* = 0.010) [133]. A recent meta-analysis including 16 studies with 727 patients showed an overall response rate to EUS-CPN of 53% (95% CI 45–62%, I^2^ 68%, *p* = 0.01) at week four, regardless of the technique (central injection, bilateral injection, or CGN). Specifically, in subgroup analysis, EUS-CGN showed the highest proportion response, with 76% (95% CI, 71–82%; I^2^ 0.01%, *p* = 0.38) and 58% (95% CI, 48–69%; I^2^ 64.9%) at week two and four, respectively [117]. Recently, a multicenter prospective trial including 51 consecutive patients [134] evaluated the effectiveness of EUS-CPN in combination with EUS-CGN, defined as a decrease in the numerical rating scale (NRS) by ≥3 points 1 week after the procedure, which was 82.4%. However, complete pain relief, defined as NRS = 0 at 1 week after the procedure, was achieved only in 27.4% of patients. 

## 5. Conclusions

In conclusion, endoscopy is an effective and safe approach for managing most of the clinical manifestations of advanced biliopancreatic tumors in the palliative setting (Figure 4). 

Furthermore, in addition to being a minimally invasive approach, which permits treating fragile and unfit-for-surgery patients, it has the advantage of treating many neoplastic clinical conditions during the same session, as in the case of MBO and mGOO, reducing anesthesiological risks and improving outcomes. However, palliative endoscopic advanced procedures require tertiary bilio-pancreatic centers due to the complexity of some techniques, mainly in the contest of EUS-guided therapeutic procedures. Moreover, tertiary centers guarantee the expertise of different specialists involved in the management of those patients with advanced bilio-pancreatic tumors, permitting them to propose the best options and manage the patient at 360 degrees, even in cases of the technical failure of endoscopic procedures.

## Figures and Tables

**Figure 1 cancers-15-05367-f001:**
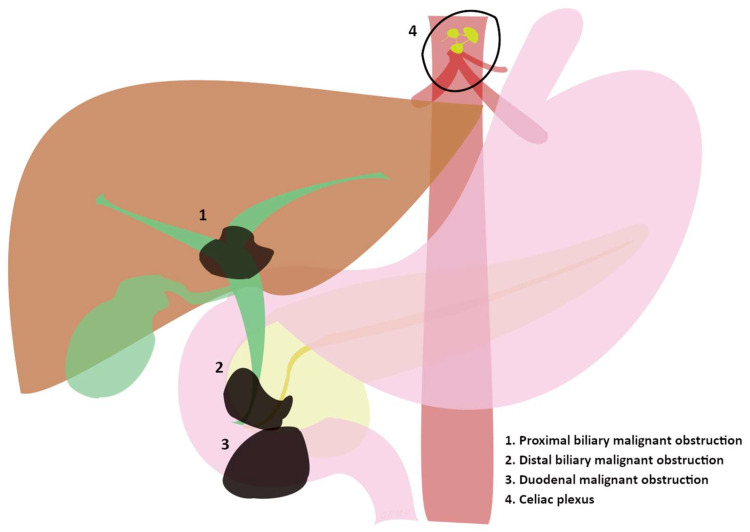
(1, 2, and 3) Sites of progression of advanced tumors involving the biliary and duodenal tract evolving in major clinical manifestations. (4) Celiac plexus as the “target” of endoscopic treatments in advanced tumors.

**Figure 2 cancers-15-05367-f002:**
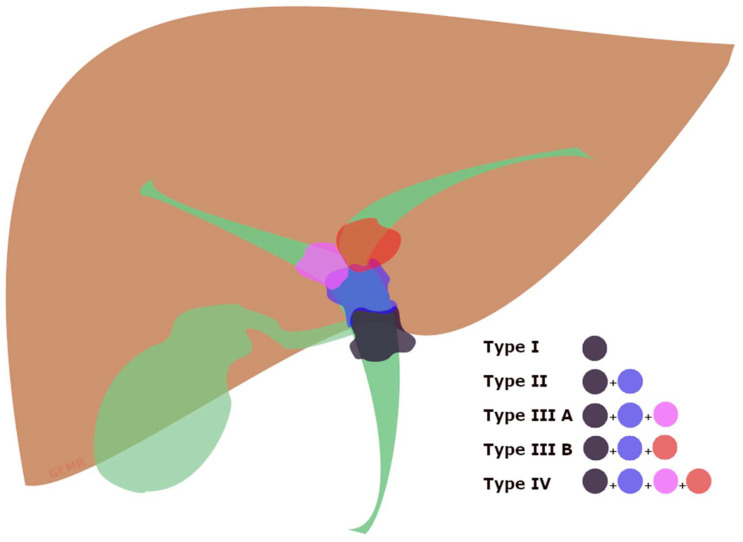
Graphical view of the Bismuth classification of Klatskin tumors [31]. Type (I) involving the common hepatic duct below the confluence; type (II) involving the biliary confluence; type (IIIA) involving the confluence and extending to the right hepatic duct; type (IIIB) involving the confluence and extending to the left hepatic duct; type (IV) involving the confluence and extending to both the right and left hepatic bile ducts.

**Figure 3 cancers-15-05367-f003:**
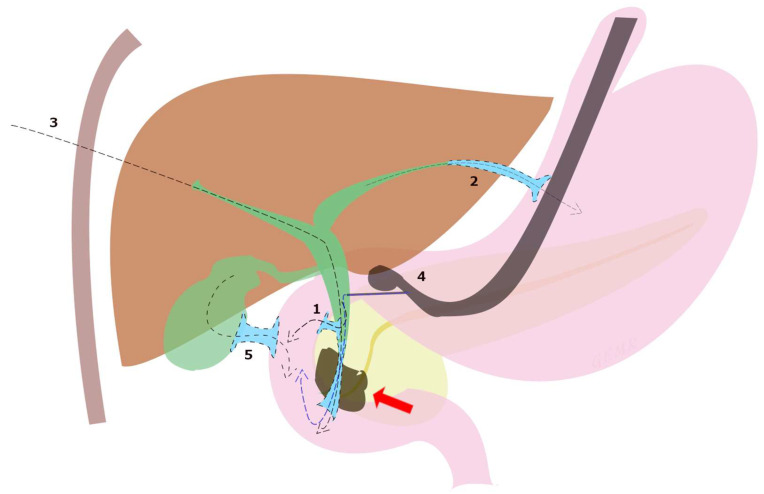
EUS-BD techniques for malignant biliary obstruction (red arrow) can be divided according to the anatomical location and the puncture site of the biliary access into (1) choledochoduodenostomy (CDS), (2) hepaticogastrostomy (HGS), (3) antegrade biliary stenting (AG), (4) rendezvous technique (RV) and (5) transduodenal gallbladder drainage (EUS-GBD).

**Figure 4 cancers-15-05367-f004:**
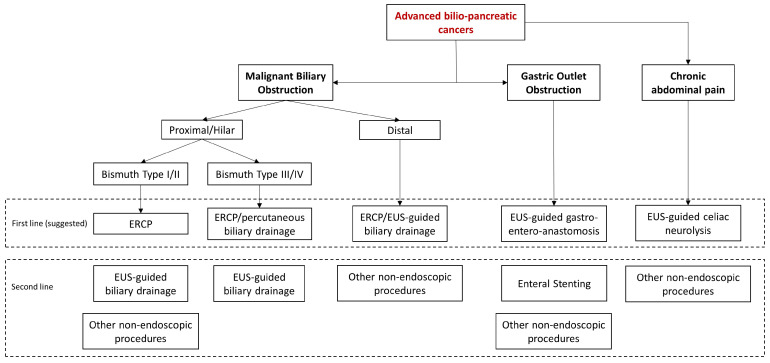
Algorithm of the management of clinical manifestations of advanced bilio-pancreatic tumors.

**Table 1 cancers-15-05367-t001:** The gastric outlet obstruction scoring system (GOOSS).

Level of Oral Intake	GOOSS Score
No oral intake	0
Liquids only	1
Soft solids	2
Low-residue or full diet	3

**Table 2 cancers-15-05367-t002:** Differences between enteral stenting, surgical GJ, and EUS-GE in patients with mGOO.

	Study Design	N° Patients	Treatments Type	Technical Success, %	Clinical Success, %	Reintervention/Recurrence Rate	Hospitalization Median (Days)	Adverse Events
Jang S., 2019 [70]	Retrospective	183 ES; 127 SGJ	ES; SGJ	96% ES; 98% SGJ	79% ES; 80% SGJ	23% ES; 23% SGJ	4 ES; 9 SGJ	6% ES; 16% SGJ
Canakis, A. 2023 [93]	Retrospective	187 EUS-GE; 123 SGJ	EUS-GE; SGJ	97.8% EUS-GE; 100% SGJ	94% EUS-GE; 94% SGJ	15.5% EUS-GE; 1.63% SGJ	NA	11.9% EUS-GE; 17.9% SGJ
Jeurnick SM, 2010 [96]	Randomized Trial	18 SGJ; 21 ES	ES; SGJ	89% SGJ; 77% ES	NA	11% SGJ; 47% ES	15 SGJ; 7 ES	33% SGJ; 47% ES
Vanella, 2023 [94]	Prospective	28 EUS-GE; 28 ES *	EUS-GE; ES	100% ES; 96,4% EUS-GE	100% ES; 100% EUS-GE	33.3% ES; 3.7% EUS-GE	7 (4–21) ES; 6.5 (3–10.5) EUS-GE	25% ES; 7.1% EUS-GE
Van Wanrooij, 2022 [97]	Retrospective	88 EUS-GE; 88 ES **	EUS-GE; ES	EUS-GE 94%; ES 98%	EUS-GE 91%; ES 75%	1% EUS-GE; 26% ES	4 (2–10.8) EUS-GE; 4 (1–9.5) ES	10% EUS-GE; 21% ES
Ge, 2019 [98]	Retrospective	78 ES; 22 EUS-GE	EUS-GE; ES	100% EUS-GE; 100% ES	95.8% EUS-GE; 76.3% ES	8.3% EUS-GE; 32% ES	mean ± SD = 7.4 (9.1) EUS-GE; 7.9 (8.2) ES	20.8% EUS-GE; 40.2% ES
Chen, 2017 [99]	Retrospective	30 EUS-GE; 52 ES	EUS-GE; ES	86.7% EUS-GE; 94.2% ES	83.3% EUS-GE; 67.3% ES	4% EUS-GE; 28.6% ES	mean ± SD = 11.3 (6.6) EUS-GE; 9.5 (8.3) ES	16.7% EUS-GE; 11.5% ES
Bronswijk, 2021 [100]	Retrospective	37 EUS-GE; 37 SGJ **	EUS-GE; SGJ	94.6% EUS-GE; 100% SGJ	97.1% EUS-GE; 89.2% SGJ	0% EUS-GE; 5.4% SGJ	4 (2–8) EUS-GE; 8 (5.5–20) EUS-GE	2.7% EUS-GE; 27% SGJ
Khashab, 2016 [101]	Retrospective	30 EUS-GE; 63 SGJ	EUS-GE; SGJ	87% EUS-GE; 100% SGJ	90% EUS-GE; 87% SGJ	3% EUS-GE; 14% SGJ	mean ± SD = 11.6 (6.6) EUS-GE; 12 (8.2) SGJ	16% EUS-GE; 25% SGJ
Perez-Miranda, 2017 [102]	Retrospective	25 EUS-GE; 29 SGJ	EUS-GE; SGJ	88% EUS-GE; 100% SGJ	84% EUS-GE; 90% SGJ	NA	mean: 9.4 EUS-GE; 8.9 SGJ	12% EUS-GE; 41% SGJ
Kouanda, 2021 [103]	Retrospective	40 EUS-GE; 26 SGJ	EUS-GE; SGJ	92.5% EUS-GE; 100% SGJ	92.5% EUS-GE; 100% SGJ	20% EUS-GE; 11.5% SGJ	5 EUS-GE; 14.5 SGJ	NA
Abbas, 2022 [104]	Retrospective	25 EUS-GE; 27 SGJ	EUS-GE; SGJ	100% EUS-GE; 100% SGJ	88% EUS-GE; 85% SGJ	NE	3.5 (2.5–9.5) EUS-GE; 9.5 (6–12) SGJ	8% EUS-GE; 41% SGJ

ES = enteral stenting; SGJ = surgical gastrojejunostomy; EUS-GE = endoscopic ultrasound-guided gastro-enterostomy; NE = not extractable. * EUS-GE and ES cohorts were matched according to baseline frailty and oncologic disease; ** after propensity score matching.

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
