# Peer review of "The Role of Endoscopy in the Palliation of Pancreatico-Biliary Cancers: Biliary Drainage, Management of Gastrointestinal Obstruction, and Role in Relief of Oncologic Pain"

_cancers, 2023, doi:10.3390/cancers15225367_

Round 1

Reviewer 1 Report

Comments and Suggestions for Authors

The review article explores the pivotal role of palliative endoscopy in managing advanced bilio-pancreatic cancers, with a focus on enhancing patients' quality of life and guiding oncologic treatments. The authors underscore the significance of both endoscopic and endoscopic ultrasound (EUS)-guided techniques, endeavoring to present a comprehensive literature review on this subject. However, I have certain concerns regarding the study's methodology and clinical reporting: 

1.     The article lacks clarity concerning the methodology employed for the literature review. Key details, such as search strategies, utilized databases, inclusion/exclusion criteria, and the assessment of articles for relevance and quality, are essential for readers to evaluate the review's validity and rigor.

2.     While the article predominantly concentrates on endoscopic interventions, it does not delve into alternative or complementary treatment approaches in chapters 3.2 and 3.3. Although the authors discuss percutaneous biliary drainage as an alternative in chapter 3.1, a critical review should encompass a comparative analysis of endoscopic techniques with other available treatments, offering a comprehensive understanding of their relative advantages, contraindications, and limitations.

3.     The article did not adequately emphasizthe potential complications and risks linked with endoscopic procedures in chapters 3.2.1, 3.2.2, and 3.3.1. A thorough critical analysis should deliberate on the balance between benefits and risks, especially within a palliative care context.

Comments on the Quality of English Language

The article could be further refined for clarity in language and structure. Improving sentence structures, avoiding repetitive phrases, can enhance the overall readability and comprehension of the content.

Author Response

The review article explores the pivotal role of palliative endoscopy in managing advanced bilio-pancreatic cancers, with a focus on enhancing patients' quality of life and guiding oncologic treatments. The authors underscore the significance of both endoscopic and endoscopic ultrasound (EUS)-guided techniques, endeavoring to present a comprehensive literature review on this subject. However, I have certain concerns regarding the study's methodology and clinical reporting: 

  1. The article lacks clarity concerning the methodology employed for the literature review. Key details, such as search strategies, utilized databases, inclusion/exclusion criteria, and the assessment of articles for relevance and quality, are essential for readers to evaluate the review's validity and rigor.

Thank you for reviewing our paper and sharing your much appreciated suggestions. We followed your indications and added a brief section in the manuscript illustrating the methodology used in our paper, and included additional materials to provide a more comprehensive explanation of our materials and methods. This is a literature review in which we also included a comprehensive search of literature, and after evaluating their relevance we included the selected studies in the sections or subsections. You’ll find the abovementioned changes in the paper and in the supplementary materials.

  1. While the article predominantly concentrates on endoscopic interventions, it does not delve into alternative or complementary treatment approaches in chapters 3.2 and 3.3. Although the authors discuss percutaneous biliary drainage as an alternative in chapter 3.1, a critical review should encompass a comparative analysis of endoscopic techniques with other available treatments, offering a comprehensive understanding of their relative advantages, contraindications, and limitations.

Based on your suggestion, we added more data and discussion on comparative analysis of endoscopic techniques with other available treatments. In consideration of the purely endoscopic scope of our work, we chose to not extensively explore alternative techniques while limiting ourselves to mentioning them in order to provide the most important data on comparison in the briefest possible way. Furthermore, we added a table to show comparative data on different treatments for MGOO.  You will find an improved and updated version of the manuscript with more comparisons among different techniques, although we did not add dedicated sections on comparisons.

  1. The article did not adequately emphasize the potential complications and risks linked with endoscopic procedures in chapters 3.2.1, 3.2.2, and 3.3.1. A thorough critical analysis should deliberate on the balance between benefits and risks, especially within a palliative care context.

Thank you for your suggestions. We agree with your point, considering the importance of balancing risks and benefits in the palliative settings of our interventions. We modified the manuscript and added data to each section, analyzing and commenting them in order to better evaluate safety.

Comments on the Quality of English Language: The article could be further refined for clarity in language and structure. Improving sentence structures, avoiding repetitive phrases, can enhance the overall readability and comprehension of the content.

We asked a native English editor to review the paper.

Reviewer 2 Report

Comments and Suggestions for Authors

This is a review article on endoscopic palliation for pancreatobiliary cancers. The paper deals with a wide range of endotherapy from biliary drainage to pain management.

1. 3.1.3. Stent selection in proximal and distal MBO differs significantly and should be discussed differently. The authos may want to move description of stent selection to 3.1.1 and 3.1.2, respectively.

2. Stent selection in neoadjuvant setting of resectable/borderline resectable pancreatic cancer is one of the topics and needs to be discussed.

3. 3.2. MGOO. The authors may want to add a table comparing enteral stenting, surgical GJ and EUS-GJ, showing advantages and disadvantages of each procedure.

4. EUS-guided ablation for pancreatic tumors has been developed and is more into clinical practice than NOTES.

5. The authors may want to change Table 1 to a figure.

6. Page 4 Line 99. ampulloma -> ampullary cancer

7. PTDB -> PTBD. Please check this throughout the paper.

Author Response

This is a review article on endoscopic palliation for pancreatobiliary cancers. The paper deals with a wide range of endotherapy from biliary drainage to pain management.

  1. 1.3. Stent selection in proximal and distal MBO differs significantly and should be discussed differently. The authors may want to move description of stent selection to 3.1.1 and 3.1.2, respectively.

Thank you for your suggestions. As requested, we divided the treatments depending on the location of the malignancy and subsequent biliary obstruction, highlighting stent selection in proximal and distal MBO.

  1. Stent selection in neoadjuvant setting of resectable/borderline resectable pancreatic cancer is one of the topics and needs to be discussed.

Considering our work is aimed at patients undergoing palliative care, we chose not to deeply cover this topic. We did however mention and comment it in paragraph 3.1.2, as you advised.

  1. 2. MGOO. The authors may want to add a table comparing enteral stenting, surgical GJ and EUS-GJ, showing advantages and disadvantages of each procedure.

Thank you for your graphical advice. We added a table with studies comparing the technical and clinical results (included safety) of the treatments of MGOO: enteral stent, surgical bypass, and EUS-guided GEA.

  1. EUS-guided ablation for pancreatic tumors has been developed and is more into clinical practice than NOTES.

Thank you for remarking this. We actually did discuss internally whether to insert a section for EUS-RFA (radiofrequency ablation) or EUS-MWA (microwave ablation) and, considering the available literature, we decided not to insert it due to its still uncommon use in daily practice for advanced bilio-pancreatic patients in the palliative care setting. In fact, even if it induces an immune-modulatory effect, which sometimes is used for combining it with neoadjuvant chemotherapy [1], they have been used as single agent or in combination with systemic chemotherapy [2] but without a controlled arm for comparing OS (overall survival) or PFS (progression-free survival). Therefore, it is still unclear whether they provide a net benefit on survival, based on current literature. A recent systematic review with meta-analysis [3] showed a high pooled clinical success, but it was defined as a “decrease in lesion size and presence of hypodense area (necrosis) on CT scan after the procedure in case of unresectable locally advanced pancreatic adenocarcinoma, metastatic pancreatic lesions, and other benign pancreatic tumors”. The latter definition does not include data on survival, so it could be useless in terms of patient life expectancy. Specifically, nowadays most indications for EUS-guided ablation include pancreatic tumors characterized as insulinoma or other symptomatic NET usually with diameter about <20 mm [4]. Therefore, we still consider it a therapy requiring more data with higher quality studies for advanced bilio-pancreatic cancers, especially with reference to long-term outcomes. We are confident it will be more used in daily clinical practice in the future and that we should focus much more on it.

  1. Scopelliti, F., et al., Technique, safety, and feasibility of EUS-guided radiofrequency ablation in unresectable pancreatic cancer. Surg Endosc, 2018. 32(9): p. 4022-4028.
  2. Oh, D., et al., Clinical outcomes of EUS-guided radiofrequency ablation for unresectable pancreatic cancer: A prospective observational study. Endosc Ultrasound, 2022. 11(1): p. 68-74.
  3. Dhaliwal, A., et al., Efficacy of EUS-RFA in pancreatic tumors: Is it ready for prime time? A systematic review and meta-analysis. Endosc Int Open, 2020. 8(10): p. E1243-E1251.
  4. Garg, R., et al., EUS-guided radiofrequency and ethanol ablation for pancreatic neuroendocrine tumors: A systematic review and meta-analysis. Endosc Ultrasound, 2022. 11(3): p. 170-185.
  5. The authors may want to change Table 1 to a figure.

Thank you for your suggestion. We moved Table 1 in the supplementary materials (Supplementary Table 1) and included a new dedicated figure in order to improve the graphic impact of the classification on the paper (Figure 2).

  1. Page 4 Line 99. ampulloma -> ampullary cancer

This change was made.

  1. PTDB -> PTBD. Please check this throughout the paper.

These changes were made.

Reviewer 3 Report

Comments and Suggestions for Authors

Interesting Paper about palliative care for advanced pancreatic or biliary cancer. In scholastic mode give an instrument to help not experiencing phisicians to choose how relieving symptoms in their patients. 
nothing of scientific or experimental only an accurate analysis of present and future techniques to increase quality of life. 

Comments on the Quality of English Language

Good and appropriate 

Author Response

Interesting Paper about palliative care for advanced pancreatic or biliary cancer. In scholastic mode give an instrument to help not experiencing physicians to choose how relieving symptoms in their patients. 
Nothing of scientific or experimental only an accurate analysis of present and future techniques to increase quality of life. 

Thank you for your evaluation and appreciation. We added supplementary materials and revised parts of the manuscript to improve it.